# G2D: From Global to Dense Radiography Representation Learning via Vision-Language Pre-training

**Che Liu**[1,2], **Cheng Ouyang**[3,8,9] **Sibo Cheng**[10]
**Anand Shah**[6,7] **Wenjia Bai**[2,3,4] **Rossella Arcucci**[1,2]

[1]Department of Earth Science and Engineering, Imperial College London, UK
[2]Data Science Institute, Imperial College London, UK
[3] Department of Computing, Imperial College London, UK
[4] Department of Brain Sciences, Imperial College London, UK
[6] Department of Infectious Disease Epidemiology, Imperial College London, UK
[7] Royal Brompton and Harefield Hospitals, UK
[8] Department of Engineering Science, University of Oxford, Oxford, UK
[9] Institute of Clinical Sciences, Imperial College London, UK
[10] CEREA, École des Ponts and EDF R&D, Île-de-France, France.
che.liu21@imperial.ac.uk

## Abstract

Medical imaging tasks require an understanding of subtle and localized visual features due to the inherently detailed and area-specific nature of pathological patterns, which are crucial for clinical diagnosis. Although recent advances in medical vision-language pre-training (VLP) enable models to learn clinically relevant visual features by leveraging both medical images and their associated radiology reports, current medical VLP methods primarily focus on aligning images with entire reports. This focus hinders the learning of dense (pixel-level) visual features and is suboptimal for dense prediction tasks (e.g., medical image segmentation). To address this challenge, we propose a novel medical VLP framework, named **G**lobal to **D**ense level representation learning (**G2D**), which aims to learn global and dense visual features simultaneously using only image-text pairs without extra annotations. In particular, G2D designs a **P**seudo **S**egmentation (**PS**) task, which enables the model to learn dense visual features during VLP. Notably, generating PS masks can be performed on the fly during VLP, which does not incur extra trainable parameters. With this simple yet effective idea, G2D achieves superior performance across 5 medical imaging tasks and 25 diseases. Particularly, in the segmentation task which requires dense visual features, G2D surpasses existing models even with just 1% of the training data for finetuning, compared to 100% used by other models. The code can be found in https://github.com/cheliu-computation/G2D-NeurIPS24/tree/main.

## 1 Introduction

In medical image analysis, learning global and dense visual representations typically requires labor-intensive and costly image and pixel-level annotations [1, 2]. Vision-language pre-training (VLP) attempts addressing this by aligning vision and language content using paired datasets [3, 4, 5, 6]. Although existing medical VLP methods excel at learning global visual features [7], they face challenges with dense visual features because the level of detail in text reports does not offer sufficient

38th Conference on Neural Information Processing Systems (NeurIPS 2024).

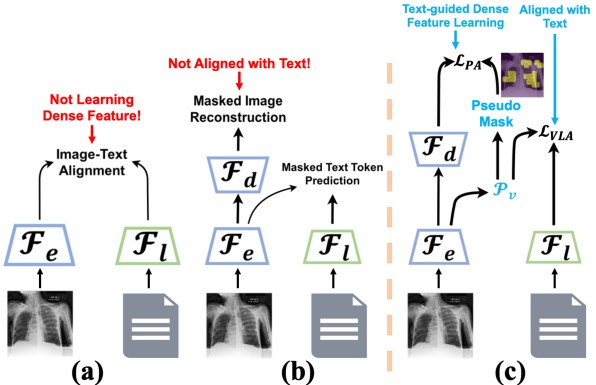

Figure 1: Comparing existing medical VLP methods with G2D: **a)** Alignment-based approaches lack dense (pixel-level) feature learning. **b)** Reconstruction-based approaches do not align with text, resulting in a deficiency in discriminative and clinically relevant visual features. **c)** The framework of **G2D (proposed)** learns dense, clinically relevant, text-aligned visual features through derived pseudo masks and image-text alignment. We use red text to highlight the deficiencies of existing methods and blue text to emphasize our advantages.

pixel-level supervision for learning these more detailed aspects. Existing medical VLP methods are categorized into two main types, as shown in Fig. 1:

- **Alignment-based Approaches**, which focus on aligning images with reports [4, 8, 9, 5, 6, 2, 10]. Although methods like [4, 8, 9] align images with entire reports and text tokens, they struggle to learn dense, clinically relevant visual features. This is due to the ambiguous supervision targets provided by text tokens, which lack explicit relational pairing with image regions, as discussed in [2].

- **Reconstruction-based Approaches**, which learn representations by reconstructing masked images or reports using masked modeling techniques [11, 12]. However, they also lack success in capturing dense, clinically relevant visual features, as the reconstruction task primarily focuses on low-level patterns (texture, shape) rather than high-level semantics [13].

Despite advancements in medical VLP, limitations still exist. Current alignment approaches align image patches with text tokens in a brute-force manner and possibly cause misalignments when some word tokens (*e.g.,* 'compatible' or 'unremarkable') lack direct visual counterparts, leading to ambiguous local alignments. Meanwhile, reconstruction-based approaches may ignore high-level image semantics. They are designed to recover low-level visual information such as intensity and texture, without accounting for high-level semantics [14, 13, 15]. As a result, both approaches perform suboptimally for downstream tasks, such as semantic segmentation and visual grounding, which require learning of granular visual features that are aligned with high-level semantics.

While numerous VLP methods are designed to capture dense visual features for natural image datasets (e.g., ImageNet) they often struggle to transfer directly to medical images because they depend on a well-trained object detection model [16, 17] or a well-aligned VLP model [18, 19]. In the medical domain, obtaining such pre-trained models is difficult as objects can be defined in various ways within a single medical image (e.g., based on organs, anatomical structures, or abnormal regions) . Additionally, in medical domain, there is a lack of foundational VLP models that are both publicly accessible and are trained on sufficiently large image-text pairs that cover diverse medical imaging applications.

In response to the aforementioned challenges, we introduce a novel medical VLP approach termed G2D. This approach is designed to extract global and dense visual representations from radiography along with their associated radiology reports, with improved **feature granularity** and enriched **semantic information**. Central to our approach is a pretext task, **P**seudo **S**egmentation (PS), which is guided by a pseudo mask (segmentation target) derived from a carefully refined and filtered attention map. PS encourages the model to learn dense representations through a pixel-level pretext task that incorporates high-level semantics. This approach, in contrast to traditional methods that align image patches with text tokens, inherently mitigates the misalignment bias and allows learning of more representative features. Notably, the PS pretext task can be implemented to run concurrently with vision-language alignment, ensuring that the model can be trained end-to-end, contrasting with the two-stage training methods [18].

To evaluate the effectiveness of G2D relative to other state-of-the-art (SOTA) VLP approaches, we deploy the pre-trained model across a diverse range of downstream tasks, including medical image classification, semantic segmentation, object detection, as well as zero-shot image classification and

visual grounding, on six public large-scale CXR datasets. The experimental results demonstrate the superior performance of G2D over existing VLP approaches on these medical applications. Overall, our contribution is three-fold:

1. We introduce G2D, the first end-to-end encoder-decoder medical VLP approach designed to learn visual representations from the global level down to the dense level, supervised by paired radiology reports and a pixel-wise pretext task.

2. We carefully design a pretext task tailored for medical VLP, pseudo segmentation. It formulates a pseudo mask as segmentation target, allowing the model to learn dense visual representations in the pretext task which can benefit downstream dense visual tasks in medicine. The pseudo mask can be generated using a parameter-free processor that leverages the attention map derived from the visual representation associated with radiology reports.

3. We conduct comprehensive experiments to validate the efficacy of the proposed G2D approach, which outperforms peer approaches across five uni-modal and cross-modal downstream tasks.

## 2  Related Works

**Alignment-based Medical VLP.** Drawing inspiration from [3], aligning images with their corresponding textual descriptions in the latent space has led to notable advancements in VLP. Within the CXR domain, while ConVIRT [4] made an early attempt at employing bidirectional contrastive learning to globally align entire images with their paired reports, there remained room for refinement. GLoRIA [8] and MGCA [9] represent advancements in image-report alignment, introducing sophisticated global-local methodologies to the field [8, 9]. These approaches endeavor to establish correspondences between distinct image and text tokens. However, it is crucial to recognize that the granularity of token-level alignment could inadvertently introduce distortions to the medical context, potentially leading to misalignments, as illustrated by [20, 2]. Med-UniC [20] utilizes augmented text in VLP training to cultivate language invariance, with the goal of mitigating linguistic biases from VLP. Meanwhile, MedKLIP [5] and KAD [21] harness domain-specific knowledge from external annotated datasets to enhance textual information extraction. Notably, these approaches [20, 5, 21] are contingent upon external resources or extra data to optimize cross-modal representation learning, which could potentially constrain their generalizability.

**Reconstruction-based Medical VLP.** Several studies, including [12, 11, 22], have employed reconstruction of image and text tokens as a pretext task within VLP. Specifically, MRM [12] endeavors to reconstruct the original image from a masked version and simultaneously aims to regenerate the original text using both the masked image and text as inputs. Conversely, PRIOR [11] adopts a strategy that focuses on cross-modal representation by reconstructing images and sentences based on complete image and report inputs. An enhancement to the MRM [12] approach is proposed by [22], where token weights are adjusted during the reconstruction phase.

While these methods have demonstrated promising outcomes, the ability of the reconstruction pretext task to capture high-level semantic representations is limited, as shown in [14, 15, 13], and is further challenged by the absence of explicit semantic-related constraints in dense visual representation learning.

## 3  Methodology

The central aim of G2D is to learn global and dense visual representations from medical images under the supervision of their corresponding radiology reports. As illustrated in Fig 2 Left, G2D integrates two alignment strategies: vision-language alignment (VLA) that learns global representations, and pixel alignment (PA) that focuses on granular representation via a pixel-level pretext task, **P**seudo **S**egmentation (**PS**). The pseudo mask for PS is constructed through a parameter-free mechanism, which is operated alongside VLA. The PS pretext task enables G2D to derive dense representations at both encoder and decoder levels during pre-training. Moreover, the task head of the pretext task facilitates a smoother transfer for the pre-trained encoder to be applied to downstream segmentation tasks, reducing the gap between the dense visual representation learned from VLP and the needs of downstream dense visual tasks after VLP. This contrasts with previous meth-

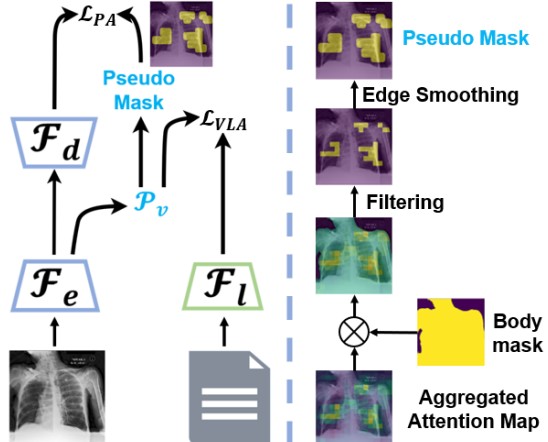

Figure 2: **Left:** Framework of G2D. **Right:** Pipeline for pseudo mask construction. We visualize the constructed pseudo mask and corresponding sentence in the radiology report in Sec A.7.

ods [4, 8, 9, 21, 5, 6] that typically transfer only the pre-trained encoder, potentially leading to an information gap between the pre-training and downstream tasks.

## 3.1 Vision-Language Contrastive Learning

We utilise a dual-encoder image-text contrastive approach following [4, 8, 9, 5]. Given a training set $S$ consisting of $N$ pairs of image-text $(v_i, l_i)$, where $v_i \in \mathcal{V}$ denotes an image and $l_i \in \mathcal{L}$ denotes a text report, $i = 1, 2, 3, ..., N$, G2D employs an image encoder $\mathcal{F}_e : \mathcal{V} \mapsto \mathbb{R}^{D_v}$ to encode the image into an embedding of dimension $D_v$, and a text encoder $\mathcal{F}_l : \mathcal{L} \mapsto \mathbb{R}^{D_l}$ to encode the text report into an embedding of dimension $D_l$. The embedded image and text features can be denoted as $\mathbf{S} = \{(\mathbf{v}_1, \mathbf{l}_1), (\mathbf{v}_2, \mathbf{l}_2), \ldots, (\mathbf{v}_N, \mathbf{l}_N)\}$, where $\mathbf{v}_i = \mathcal{F}_e(v_i)$ and $\mathbf{l}_i = \mathcal{F}_l(l_i)$.

As depicted in Fig. 2, G2D incorporates two alignment strategies: VLA and PA. For VLA, the model aims to learn global visual and text representations by pulling the embeddings of paired image-report samples closer, while distancing embeddings of unpaired samples, using a contrastive loss $\mathcal{L}_{\text{VLA}}$. The objective of contrastive learning is to predict $N$ positive matched pairs $(v_i, l_i)$ and $N^2 - N$ negative pairs among $N \times N$ possible image-text pair combinations [3]. Subsequently, two non-linear vision and language projectors $\mathcal{P}_v$ and $\mathcal{P}_l$ transform $\mathbf{v}_i$ and $\mathbf{l}_i$ into the same dimension $d$, where $\hat{\mathbf{v}}_i = \mathcal{P}_v(\mathbf{v}_i)$, $\hat{\mathbf{l}}_i = \mathcal{P}_l(\mathbf{l}_i)$, and $\hat{\mathbf{v}}_i, \hat{\mathbf{l}}_i \in \mathbb{R}^d$. After obtaining image feature vectors $[\hat{\mathbf{v}}_i]_{i=1}^N$ and text feature vectors $[\hat{\mathbf{l}}_i]_{i=1}^N$ with the same dimension $d$, the contrastive loss $\mathcal{L}_{\text{VLA}}$ can be formulated as:

$$\mathcal{L}_{\text{VLA}} = -\frac{1}{K} \sum_{i=1}^{N} \left( \log \frac{\exp(\hat{\mathbf{v}}_i^\top \hat{\mathbf{l}}_i / \sigma)}{\sum_{j=1}^{K} \exp(\hat{\mathbf{v}}_i^\top \hat{\mathbf{l}}_j / \sigma)} \right) \tag{1}$$

$\sigma$ denotes the temperature hyper-parameter empirically set to 0.07 following [9], and $K \in N$ is the batch size.

## 3.2 Pseudo Segmentation Mask Construction

Notably, although MedSAM [23] claims to build image-mask pairs, it requires box prompt inputs not available in the MIMIC-CXR [24] dataset. Designing a box prompt for each image is labor-intensive and unfeasible for this work, so we construct the pseudo mask based on attention maps.

**Attention Aggregation.** Inspired by CLIP [3], we incorporate an attention pooling mechanism in conjunction with the non-linear projector $\mathcal{P}_v$ to derive a pixel-wise attention map. A dense feature map $\mathbf{V}_i$ is extracted from the final convolutional layer before the pooling operation in the image encoder $\mathcal{F}e$, with the dimension $C \times H \times W$. Here, $C$ denotes the number of channels, while $H$ and $W$ represent the height and width of the feature maps. Subsequently, we reshape $\mathbf{V}_i$ into a dimension of $HW \times C$. In this way, $\mathbf{V}_i$ can be interpreted as a sequence of pixel embeddings, where each token in this sequence represents the embedding of an individual pixel. The length of this sequence is defined by the number of channels, $C$. A special token, [CLS], is introduced to aggregate all pixel embeddings through multi-head self-attention (MHSA) [25, 3]. This process offers an attention score matrix $W_i^h$ for each pixel, with dimensions $h \times H \times W$. Here, $h$ signifies the

attention head number, and $h \in \mathcal{H}$, with $\mathcal{H}$ being the total number of attention heads. This attention score matrix characterizes the information exchange between pixels and semantics provided by the text [3, 18], and therefore it carries semantic information and is an ideal candidate for constructing the pretext pseudo mask. To derive the pseudo mask, we aggregate $W_i^h$ across all attention heads to produce $\hat{W}_i$, as described by:

$$\hat{W}_i = \frac{\sum_{h=1}^{\mathcal{H}} W_i^h}{h} \tag{2}$$

**Mask Filtering and Edge Smoothing.** After obtaining the aggregated attention map $\hat{W}_i$, we up-sample it to match the original image dimensions $H' \times W'$. To remove pseudo mask regions in the background, we construct a body mask for each CXR image using a histogram-based thresholding approach, following common practice [26, 27]. Subsequently, all attention scores outside the body mask are set to zero. A threshold is applied to filter out low attention scores within the body mask, transforming $\hat{W}_i$ into a binary mask. The threshold is determined at the $85\%$ percentile of attention scores from $\hat{W}_i$. The binary pseudo mask $M_i$ is formulated as:

$$M_i^{j,k} = \begin{cases} 1 & \text{if } W_i^{j,k} \geq \text{threshold} \\ 0 & \text{otherwise} \end{cases}, \text{where}$$

$$j = 1, 2, 3, ..., H', k = 1, 2, 3, ..., W' \tag{3}$$

To smooth the square-like boundary in the mask caused by upsampling, we apply bilateral filtering (BF) [28] to $M_i$, resulting in a refined pseudo mask $\tilde{M}_i$, as shown in Fig. 2 Right. A comprehensive ablation study discussing the threshold and smoothing operation is presented in Sec. 4.5.

## 3.3 Dense Visual Representation Learning through Pseudo Segmentation in VLP

While the global visual representation can be learned via VLA, dense representation often lacks direct alignment. To tackle this limitation, we introduce an image decoder, denoted as $\mathcal{F}_d$, as shown in Fig. 2 Left. This decoder takes visual feature $\mathbf{V}_i$ as input and utilises the pseudo mask $\tilde{M}_i$ as the supervisory signal for the pretext task. We employ the commonly used soft Dice loss and binary cross-entropy loss [27] to optimise this task. The training loss function for $\mathcal{L}_{\text{PA}}$ is formulated as:

$$\mathcal{L}_{\text{PA}} = \frac{1}{2}(\mathcal{L}_{\text{Dice}} + \mathcal{L}_{\text{BCE}}),$$

$$\mathcal{L}_{\text{Dice}} = \sum_{i=1}^{K} \sum_{j=1}^{H'} \sum_{k=1}^{W'} \left( 1 - \frac{2 \times (\tilde{M}'_{i,j,k} \odot \tilde{M}_{i,j,k})}{\tilde{M}'_{i,j,k} + \tilde{M}_{i,j,k}} \right),$$

$$\mathcal{L}_{\text{BCE}} = -\sum_{i=1}^{K} \sum_{j=1}^{H'} \sum_{k=1}^{W'} \left[ \tilde{M}_{i,j,k} \log(\tilde{M}'_{i,j,k}) + (1 - \tilde{M}_{i,j,k}) \log(1 - \tilde{M}'_{i,j,k}) \right],$$

$$\text{with } \tilde{M}'_i = \mathcal{F}_d(\mathbf{V}_i) \tag{4}$$

The total loss for G2D is the sum of the VLA loss (Eq. 1) and the PA loss (Eq. 4):

$$\mathcal{L}_{total} = \mathcal{L}_{\text{VLA}} + \mathcal{L}_{\text{PA}} \tag{5}$$

It is worth noting that the pseudo mask is designed as a pixel-wise pretext supervisory signal. Although there is no manual annotation involved, the pseudo mask is constructed from the visual feature of the image encoder, which is pre-trained to align with radiology reports and thus contains clinical knowledge such as anatomical regions mentioned by the reports. In this sense, it can be a good surrogate target for learning pixel-wise semantic information. To demonstrate that the pseudo mask serves as a meaningful target for dense visual pre-training, we conduct an ablation study to use a perturbed pseudo mask with corrupt semantics for pre-training, and compare it to the proposed pseudo mask, as detailed in Table 8 and Sec A.6.

# 4 Experiments and Analysis

In this section, we compare our approach with SOTA medical VLP techniques. The implementation details and dataset training/test splits are reported in Sec A.3, A.4.

**Pretraining Dataset and Configuration** We utilise the MIMIC-CXR dataset [29, 24]. After preprocessing based on established protocols [9, 5], it provides 213,384 image-text pairs for pre-training. For the VLP part, we employ a standard ResNet-50 as the vision encoder $\mathcal{F}_e$ and adopt the decoder part of a U-Net as the vision decoder $\mathcal{F}_d$. We adopt ClinicalBERT [30] as the text encoder using configurations described in [5, 21]. In line with [9, 8], G2D is pre-trained for 50 epochs across 16 A100 GPUs, each accommodating a batch size of 128. The AdamW optimizer is employed with a learning rate set to $2 \times 10^{-4}$ and a weight decay of $1 \times 10^{-8}$. Additionally, a linear warm-up and a cosine annealing scheduler are incorporated in the training process.

## 4.1 Downstream Task Datasets and Configurations

For downstream tasks, our focus is to evaluate the efficacy of G2D in learning granular visual features that can be used for localisation, vision-language understanding, and visual recognition tasks. We examine the capability and transferability of the learned cross-modal representations by using them for five distinct medical imaging tasks, covering a spectrum of 25 different diseases.

**Medical Image Segmentation.** This task utilises the RSNA [31] and SIIM [32] datasets, following preprocessing guidelines established in [9, 8]. We adopt U-Net [1] fine-tuning configurations following [8, 9]. The pre-trained vision encoder is frozen, while only the decoder parameters are updated during fine-tuning. Performance is assessed using the Dice score, following the evaluation protocol in [8, 9]. It is noteworthy that the original MedKLIP [5] uses a different configuration (***updating the vision encoder***) compared to other methods (***freezing the vision encoder***) [4, 8, 9, 6]. Therefore, in these experiments, we reference the results reported in [20], which reimplemented MedKLIP under a setting consistent with all other methods. For a fair comparison specifically with MedKLIP, we also reimplement G2D under MedKLIP's original setting, as reported in the Sec A.5.

**Medical Object Detection.** This task is conducted using the RSNA dataset [31] for Pneumonia Detection and the Object-CXR dataset [33] for Foreign Objects Detection, adhering to preprocessing methods from [9]. We employ YOLOv3 [34] for detection, using the pre-trained vision encoder and updating an additional detection head during fine-tuning. We report the mean Average Precision (mAP) with IoU thresholds between 0.4∼0.75. The setup for this task is in accordance with in [9].

**Zero-shot Medical Image Visual Grounding.** In accordance with [5], this task is conducted on the RSNA [31] and SIIM [32] datasets, using the same official data split and evaluation metrics. We employ CXR images as input and utilise the corresponding ground truth label maps for assessing the grounding performance, in terms of recall, IoU, and Dice score.

**Zero-shot Medical Image Classification.** In compliance with the guidelines set forth in [5, 21], we conduct this task on the RSNA [31], SIIM [32], CheXpert [35], and CXR14 [36] datasets. For the RSNA and SIIM datasets, we employ the test set splits provided by MedKLIP [5], given that KAD [21] did not conduct experiments on these two datasets. For the CheXpert and CXR14 datasets [35, 36], we use the official test set splits to ensure a fair comparison with KAD [21]. It is important to note that MedKLIP [5] creates its own test split rather than using the official test split. Hence, we do not use MedKLIP's splits in our experiments. We report the results using the macro average of AUC, F1, and ACC scores across all diseases.

**Medical Image Fine-tuned Classification.** In alignment with [5, 21], we use the CXR14 dataset [36], comprising 112,120 frontal-view X-rays from 30,805 patients, annotated for 14 diseases. We adhere to the official split for consistent evaluation, following KAD [21]. It is worth noting that MedKLIP does not use the official data split. Hence, we refer to the results reported in KAD [21] rather than those from the original MedKLIP [5]. To ensure a fair comparison with MedKLIP, we reimplemented G2D for this experiment under the MedKLIP configuration, as detailed in Sec A.5. CXR images are resized to $256 \times 256$ [21]. During fine-tuning, all model parameters are updated, including the pre-trained vision encoder and linear classifier. The AdamW optimizer is used with a learning rate of $1 \times 10^{-4}$ and a batch size of 64 for 50 epochs. Evaluation is based on the AUC score, adhering to the protocol outlined in [8, 9, 12].

Table 1: Results of semantic segmentation and object detection. Best results are highlighted in bold, with '-' denoting mAP values $< 1\%$. Methods with $\star$ use disease-level annotations. '/' indicates object detection not deployable with encoder-decoder architecture. The MedKLIP results in this table differ from the original work [5] because MedKLIP fine-tuned the encoder in their original study, whereas other methods froze the encoder. To ensure fairness, we reimplemented MedKLIP with the frozen encoder for comparison in this table. Additionally, for a fair comparison specifically with MedKLIP, we compare G2D with MedKLIP under its original configuration in Tab 7 and Sec A.5.

| Tasks | Semantic Segmentation (Dice) | | | | | | Object Detection (mAP) | | | | | |
|---|---|---|---|---|---|---|---|---|---|---|---|---|
| **Datasets** | **SIIM** | | | **RSNA** | | | **RSNA** | | | **Object CXR** | | |
| **Methods** | 1% | 10% | 100% | 1% | 10% | 100% | 1% | 10% | 100% | 1% | 10% | 100% |
| Random Init | 9.0 | 28.6 | 54.3 | 6.9 | 10.6 | 18.5 | 1.0 | 4.0 | 8.9 | - | 0.5 | 4.4 |
| ImageNet Init | 10.2 | 35.5 | 63.5 | 34.8 | 39.9 | 64.0 | 3.6 | 8.0 | 15.7 | - | 2.9 | 8.3 |
| ConVIRT [4] | 25.0 | 43.2 | 59.9 | 55.0 | 67.4 | 67.5 | 8.2 | 15.6 | 17.9 | - | 8.6 | 15.9 |
| GLoRA [8] | 35.8 | 46.9 | 63.4 | 59.3 | 67.5 | 67.8 | 9.8 | 14.8 | 18.8 | - | 10.6 | 15.6 |
| GLoRIA-MIMIC [8] | 37.4 | 57.1 | 64.0 | 60.3 | 68.7 | 68.3 | 11.6 | 16.1 | 24.8 | - | 8.90 | 16.6 |
| MGCA [9] | 49.7 | 59.3 | 64.2 | 63.0 | 68.3 | 69.8 | 12.9 | 16.8 | 24.9 | - | 12.1 | 19.2 |
| M-FLAG [6] | 52.5 | 61.2 | 64.8 | 64.6 | 69.7 | 70.5 | 13.7 | 17.5 | 25.4 | - | 12.4 | 19.3 |
| MedKLIP* [5] | 50.2 | 60.8 | 63.9 | 66.2 | 69.4 | 71.9 | 8.9 | 16.3 | 24.5 | - | 7.1 | 11.6 |
| **Ours (encoder)** | **62.6** | **63.1** | **66.8** | **70.9** | **72.6** | **75.1** | **15.9** | **21.7** | **27.2** | **3.8** | **13.1** | **20.4** |
| **Ours (encoder-decoder)** | **65.6** | **66.9** | **68.4** | **72.8** | **73.4** | **76.9** | / | | | | | |

**Medical Image Linear Classification.** In strict accordance with the configuration in [8, 4, 9], this task is conducted on the CheXpert [35], RSNA [31], and COVIDx [37] datasets. We only update a randomly initialized linear classification layer, while the pre-trained vision encoder remains frozen. For fair evaluation, we employ AUC scores on CheXpert and RSNA, along with accuracy metrics on COVIDx, as mentioned in [8, 9]. Apart from zero-shot image classification and visual grounding, we fine-tune using $1\%, 10\%, 100\%$ of the training data for all downstream tasks. Detailed settings, including implementation and data splits, are outlined in Sec A.4.

## 4.2 Performance on Visual Localisation Tasks

In Tab 1, following [16, 38], we evaluate G2D alongside other SOTA approaches on two pivotal visual localisation tasks: semantic segmentation and object detection. The aim is to assess the efficacy of the dense visual features learned.

Initially, we transfer only the encoder weights from the pre-trained G2D for the segmentation task, adhering to the protocols of [9, 8, 4, 6]. In this setup, our approach consistently achieves the highest performance across all data fractions for both SIIM [32] and RSNA datasets [31]. To assess the impact of the visual decoder pre-trained with the PS pretext task, we transfer the weights of both the encoder and decoder from G2D for the segmentation task, resulting in striking outcomes. Remarkably, with just 1% of training data, G2D surpasses the performance of all peer methods, even those trained with a full 100% of training data. This observation underlines the fact that the pixel-level pretext task, PS, significantly improves the quality of dense visual features derived from VLP, which provide advantages for the downstream segmentation task.

In object detection, our method consistently outperforms existing methods across all data fractions for both RSNA and Object-CXR datasets [31, 33]. Notably, G2D achieves a 3.8% mAP on the Object-CXR dataset with just 1% of the data for fine-tuning, a significant leap from other methods that scarcely reach a 1% mAP.

These results highlight the efficacy of our proposed model, G2D, and the pretext task, PS, especially in semantic segmentation tasks that rely on dense visual features. PS not only enables G2D to learn visual representations in the encoder-decoder structure but also reduces the gap between pre-training and downstream tasks. By enhancing the encoder's ability to capture global and dense features simultaneously, PS surpasses existing approaches, proving particularly advantageous for object detection tasks that heavily rely on dense features [39].

## 4.3 Performance on Vision-Language Understanding

In Tab 2, we evaluate the efficacy of G2D on vision-language understanding tasks, zero-shot visual grounding and zero-shot image classification. For the zero-shot visual grounding task, our proposed method outperforms peer approaches. Specifically, on the SIIM dataset [32], it achieves a leading Dice score of 5.1. This dominance persists in the RSNA dataset [31], where our method reaches a

Table 2: Comparison between G2D (ours) and various other medical VLP methods in vision-language understanding tasks, with the best results emphasized in bold. Methods marked with ⋆ utilize extra annotated data during pre-training. '/' indicates that the original work did not report the results. Notably, KAD [21] does not report ACC for the CheXpert dataset.

(a) Results of zero-shot visual grounding task.

| Task | Zero-shot Visual Grounding | | | | | |
|---|---|---|---|---|---|---|
| Datasets Methods | SIIM | | | RSNA | | |
| | Recall | IoU | Dice | Recall | IoU | Dice |
| GLoRIA [8] | 23.8 | 1.2 | 2.1 | 83.3 | 21.8 | 34.7 |
| BioViL [40] | 19.6 | 1.7 | 2.6 | 85.2 | 30.3 | 43.9 |
| MedKLIP* [5] | 35.6 | 2.1 | 4.0 | 86.6 | 31.7 | 46.5 |
| **Ours** | **37.7** | **3.9** | **5.1** | **88.4** | **33.5** | **47.7** |

(b) Results of zero-shot image classification task.

| Task | Zero-shot Image Classification | | | | | | | | | | |
|---|---|---|---|---|---|---|---|---|---|---|---|
| Datasets Methods | RSNA | | | SIIM | | | CXR14 | | | CheXpert | |
| | AUC | F1 | ACC | AUC | F1 | ACC | AUC | F1 | ACC | AUC | F1 |
| ConVIRT [4] | 80.4 | 58.4 | 76.1 | 64.3 | 43.3 | 57.0 | 56.0 | 13.5 | 45.9 | 59.0 | 26.4 |
| GLoRIA [8] | 71.5 | 49.0 | 71.3 | 53.4 | 38.2 | 40.5 | 61.0 | 17.4 | 50.3 | 75.0 | 57.0 |
| BioViL [40] | 82.8 | 58.3 | 76.7 | 70.8 | 48.6 | 69.1 | 66.2 | 66.2 | 63.3 | 69.3 | 46.3 |
| CheXzero* [5] | 85.8 | 62.1 | 79.4 | 68.8 | 47.0 | 54.7 | / | / | / | 88.9 | 60.6 |
| MedKLIP* [5] | 86.9 | 63.4 | 80.0 | 89.2 | 68.3 | 84.3 | 72.6 | 24.4 | 79.6 | 87.9 | 61.4 |
| KAD* [21] | / | / | / | / | / | / | 78.9 | 32.3 | 81.6 | 90.5 | 64.6 |
| **Ours** | **87.6** | **64.8** | **81.5** | **89.7** | **69.3** | **85.4** | **79.4** | **33.1** | **82.3** | **91.2** | **65.6** |

Table 3: Evaluation of image classification fine-tuning on the CXR14 dataset is conducted, with all metrics presented as AUC scores, where the mean metric is macro-averaged. Best performances are highlighted in bold. Methods marked with ⋆ utilize extra annotated data for pre-training. MedKLIP's results here differ from the original study [5] as it did not utilize the official test split, unlike KAD [21]. We use the result of MedKLIP reported by KAD [21], which reimplemented MedKLIP on the official test set for fairness. All results in this table are sourced from KAD [21]. To compare fairly with MedKLIP, we assess G2D against its original configuration in Tab 7 and Sec A.5.

| Data fraction | Method | Mean | Atelectasis | Cardiomegaly | Effusion | Infiltration | Mass | Nodule | Pneumonia | Pneumothorax | Consolidation | Edema | Emphysema | Fibrosis | Pleural Thicken | Hernia |
|---|---|---|---|---|---|---|---|---|---|---|---|---|---|---|---|---|
| 1% | Random Init | 58.1 | 55.7 | 57.7 | 63.6 | 61.6 | 55.0 | 60.2 | 57.1 | 58.2 | 60.8 | 63.3 | 53.4 | 63.7 | 56.8 | 46.0 |
| | ImageNet Init | 63.5 | 66.2 | 64.2 | 72.1 | 57.0 | 59.0 | 58.5 | 60.0 | 62.6 | 62.4 | 66.8 | 61.5 | 70.7 | 63.1 | 64.5 |
| | ConVIRT [4] | 64.9 | 66.0 | 78.2 | 78.9 | 61.1 | 59.6 | 65.5 | 60.8 | 65.7 | 60.5 | 60.7 | 65.8 | 68.0 | 62.7 | 46.6 |
| | GLoRIA [8] | 59.7 | 59.7 | 56.7 | 74.1 | 64.6 | 55.9 | 55.7 | 61.1 | 60.7 | 66.5 | 66.9 | 55.0 | 55.8 | 59.2 | 43.6 |
| | BioViL [40] | 57.9 | 55.5 | 56.4 | 72.2 | 65.0 | 56.7 | 54.6 | 62.6 | 56.0 | 65.7 | 68.1 | 51.6 | 51.3 | 59.2 | 36.0 |
| | MedKLIP* [5] | 60.9 | 65.5 | 59.0 | 74.5 | 64.3 | 55.0 | 61.1 | 60.9 | 59.9 | 65.9 | 68.2 | 53.5 | 64.8 | 59.3 | 40.0 |
| | KAD* [21] | 78.7 | 77.0 | 88.2 | 82.9 | 69.2 | 75.1 | 69.7 | 73.5 | 86.1 | 72.7 | 81.3 | 89.3 | 74.3 | 69.2 | 93.8 |
| | **Ours** | **79.1** | **78.1** | **88.3** | **83.1** | **70.2** | **75.4** | **69.7** | **74.0** | **86.5** | **72.9** | **81.6** | **90.2** | **74.4** | **69.5** | **94.1** |
| 10% | Random Init | 69.1 | 68.2 | 76.6 | 74.6 | 67.4 | 62.3 | 58.0 | 63.6 | 72.8 | 67.8 | 78.0 | 64.7 | 71.5 | 65.3 | 77.1 |
| | ImageNet Init | 72.6 | 70.9 | 79.8 | 76.9 | 68.4 | 69.3 | 65.6 | 63.0 | 79.3 | 67.1 | 76.7 | 74.9 | 72.9 | 71.1 | 81.0 |
| | ConVIRT [4] | 77.1 | 74.0 | 84.3 | 81.1 | 69.3 | 74.8 | 70.0 | 67.1 | 82.8 | 70.1 | 81.4 | 87.1 | 76.7 | 71.9 | 89.3 |
| | GLoRIA [8] | 74.3 | 72.1 | 80.8 | 80.0 | 68.7 | 73.3 | 67.5 | 65.8 | 77.9 | 67.6 | 79.7 | 79.9 | 78.7 | 69.3 | 78.7 |
| | BioViL [40] | 72.7 | 70.3 | 78.5 | 79.0 | 66.6 | 71.8 | 67.1 | 66.5 | 76.7 | 68.4 | 79.9 | 76.1 | 74.8 | 65.3 | 76.3 |
| | MedKLIP* [5] | 74.8 | 72.9 | 80.2 | 79.3 | 69.8 | 71.9 | 68.1 | 66.6 | 79.6 | 69.6 | 81.1 | 79.5 | 75.6 | 71.3 | 81.9 |
| | KAD* [21] | 80.7 | 77.6 | 88.9 | 83.3 | 71.8 | 78.3 | 71.9 | 73.7 | 87.2 | 75.0 | 83.3 | 90.3 | 80.7 | 72.3 | 95.3 |
| | **Ours** | **81.1** | **78.4** | **89.3** | **83.7** | **72.2** | **78.8** | **72.3** | **74.1** | **87.8** | **75.3** | **84.0** | **90.4** | **80.8** | **72.5** | **95.4** |
| 100% | Random Init | 79.0 | 75.0 | 87.9 | 81.5 | 69.1 | 79.8 | 72.6 | 70.3 | 82.6 | 73.1 | 83.9 | 83.5 | 80.7 | 75.4 | 90.3 |
| | ImageNet Init | 80.4 | 76.3 | 86.7 | 82.3 | 69.3 | 82.3 | 76.3 | 71.9 | 84.0 | 73.7 | 84.2 | 89.3 | 81.9 | 77.0 | 89.9 |
| | ConVIRT [4] | 80.8 | 77.1 | 86.7 | 82.5 | 70.3 | 81.8 | 76.1 | 72.2 | 85.7 | 74.7 | 85.4 | 90.1 | 80.9 | 77.1 | 90.9 |
| | GLoRIA [8] | 80.0 | 76.0 | 85.5 | 81.8 | 70.0 | 81.4 | 74.9 | 71.5 | 82.8 | 73.9 | 83.2 | 88.7 | 81.3 | 76.7 | 92.1 |
| | BioViL [40] | 80.0 | 76.5 | 87.1 | 82.4 | 69.7 | 81.9 | 75.2 | 71.0 | 84.5 | 74.2 | 84.2 | 87.1 | 82.1 | 75.9 | 88.8 |
| | MedKLIP* [5] | 80.1 | 76.4 | 84.9 | 82.3 | 69.7 | 82.0 | 74.7 | 71.2 | 83.9 | 75.1 | 84.8 | 87.9 | 81.7 | 77.7 | 89.2 |
| | KAD* [21] | 82.5 | 78.5 | 89.7 | 84.0 | 71.3 | 83.6 | 77.1 | 74.0 | 87.4 | 75.3 | 86.0 | 91.6 | 82.9 | 77.8 | 96.1 |
| | **Ours** | **83.1** | **79.9** | **90.2** | **84.5** | **71.8** | **84.2** | **78.0** | **74.2** | **87.7** | **75.6** | **86.9** | **92.0** | **83.1** | **78.2** | **96.5** |

Dice score of 47.7, surpassing other SOTA approaches. When examining zero-shot image classification, our method again shows its superiority across the AUC, F1, and ACC metrics on both the RSNA [31] and SIIM datasets [32]. Such consistent and superior outcomes underscore the adaptability and effectiveness of G2D in handling vision-language understanding tasks, indicating that integrating PS into G2D can enhance not only uni-modal but also cross-modal tasks.

## 4.4 Performance on Visual Recognition Tasks

In our final assessment focused on visual recognition, Tab 3 demonstrates our method's consistent supremacy on the CXR14 dataset [36] for fine-tuned disease classification across 1%, 10%, and 100% training data. Similarly, Tab 4 underscores that G2D achieves the highest performance on the CheXpert, RSNA, and COVIDx datasets [35, 31, 37] for linear evaluation across all training data ratio. Notably, G2D consistently outperforms even those methods like MedKLIP and KAD [41] that leverage additional disease-level annotations during pre-training stage. This demonstrates G2D's representative visual features, suggesting that enhancing dense representation learning via PS can also improve results in tasks primarily anchored on global representation.

Table 4: Linear classification results for CheXpert, RSNA, and COVIDx datasets with 1%, 10%, and 100% training data. The best results are highlighted in bold. Methods with ⋆ leverage disease-level annotations for pre-training. The evaluation metric follows [9].

| Datasets (Metric) | CheXpert (AUC) | | | RSNA (AUC) | | | COVIDx (ACC) | | |
|---|---|---|---|---|---|---|---|---|---|
| Methods | 1% | 10% | 100% | 1% | 10% | 100% | 1% | 10% | 100% |
| Random Init | 56.1 | 62.6 | 65.7 | 58.9 | 69.4 | 74.1 | 50.5 | 60.3 | 70.0 |
| ImageNet Init | 74.4 | 79.7 | 81.4 | 74.9 | 74.5 | 76.3 | 64.8 | 78.8 | 86.3 |
| ConVIRT [4] | 85.9 | 86.8 | 87.3 | 77.4 | 80.1 | 81.3 | 72.5 | 82.5 | 92.0 |
| GLoRIA [8] | 86.6 | 87.8 | 88.1 | 86.1 | 88.0 | 88.6 | 67.3 | 77.8 | 89.0 |
| GLoRIA-MIMIC [8] | 87.1 | 88.7 | 88.0 | 87.0 | 89.4 | 90.2 | 66.5 | 80.5 | 88.8 |
| MGCA [9] | 87.6 | 88.0 | 88.2 | 88.6 | 89.1 | 89.9 | 72.0 | 83.5 | 90.5 |
| MRM [12] | 88.5 | 88.5 | 88.7 | 91.3 | 92.7 | 93.3 | 66.9 | 79.3 | 90.8 |
| MedKLIP⋆ [5] | 86.2 | 86.5 | 87.7 | 87.3 | 88.0 | 89.3 | 74.5 | 85.2 | 90.3 |
| **Ours** | **89.7** | **90.4** | **91.1** | **92.2** | **92.9** | **93.6** | **76.6** | **88.2** | **93.4** |

Table 5: Results of various ablation experiments. The best results are bolded.

(a) Loss for the decoder. 'None' indicates Encoder-Only visual backbones.

| Decoder Loss | SIIM Dice | RSNA mAP | CXR14 AUC |
|---|---|---|---|
| None | 49.2±1.5 | 11.7±1.2 | 77.1±1.5 |
| Reconstruction | 53.4±1.3 | 13.0±0.9 | 77.3±2.1 |
| **Pseudo Seg (Ours)** | **65.6±1.7** | **15.9±0.8** | **79.1±1.2** |

(b) Threshold for constructing pseudo segmentation masks.

| Threshold | SIIM Dice | RSNA mAP | CXR14 AUC |
|---|---|---|---|
| **85% percentile** | **65.6±1.7** | **15.9±0.8** | **79.1±1.2** |
| 75% percentile | 63.0±2.1 | 14.1±1.2 | 78.3±2.0 |
| median | 58.8±1.6 | 12.5±2.3 | 75.6±1.1 |
| GMM [42] | 59.2±1.5 | 12.9±1.4 | 75.2±1.9 |

(c) Ablation of the number of dimensions of projectors.

| Num of Dim | SIIM Dice | RSNA mAP | CXR14 AUC |
|---|---|---|---|
| 128 | **65.6±1.7** | 15.9±0.8 | **79.1±1.2** |
| 256 | 64.9±1.9 | **16.1±1.1** | 78.3±1.5 |
| 512 | 64.6±1.2 | 15.7±1.0 | 78.0±1.3 |

(d) Ablation of multi-head attention maps aggregation.

| Method | SIIM Dice | RSNA mAP | CXR14 AUC |
|---|---|---|---|
| **w Aggregation** | **65.6±1.7** | **15.9±0.8** | **79.1±1.2** |
| w/o Aggregation | 62.1±2.2 | 13.5±1.7 | 77.5±2.3 |

(e) Number of attention heads.

| Heads | SIIM Dice | RSNA mAP | CXR14 AUC |
|---|---|---|---|
| 1 | 63.4±2.0 | 14.2±1.4 | 78.2±1.0 |
| 2 | 64.7±1.6 | 15.1±2.3 | 78.8±1.5 |
| **3** | **65.6±1.7** | **15.9±0.8** | **79.1±1.2** |
| 4 | 65.3±1.6 | 15.4±0.9 | 78.7±1.9 |

(f) Refinement of Pseudo Segmentation Masks

| | SIIM Dice | RSNA mAP | CXR14 AUC |
|---|---|---|---|
| w/o body mask | 63.4±1.5 | 15.3±2.1 | 78.4±1.6 |
| w/o edge smoothing | 64.1±1.2 | 15.2±1.7 | 78.5±2.2 |
| w both (Ours) | **65.6±1.7** | **15.9±0.8** | **79.1±1.2** |

## 4.5 Ablation Studies

**Pseudo Segmentation vs. Reconstruction.** In Tab 5a, we evaluate the impact of the proposed PS pretext task in comparison to pixel reconstruction and models without a decoder-level constraint. The model pre-trained with PS outperforms the other two approaches across all three downstream tasks, particularly in semantic segmentation. While the model pre-trained with a pixel reconstruction constraint exhibit improved performance compared to unconstrained variants, such models still underperform the model with the PS constraint. These results underscore the effectiveness of decoder-level pretext tasks and suggest that an emphasis on high-level semantics, derived from PS, is more beneficial than focusing on the low-level semantics from pixel reconstruction. The PS potentially reduces the disparity between features learned through VLP and those required by downstream semantic segmentation tasks. It also enables the model to acquire more representative features that are beneficial for various tasks.

**Threshold of Pseudo Mask Construction.** As shown in Tab 5b, performance varies with different thresholds, with the 85% percentile threshold proving most effective across all three downstream tasks. Despite employing the Gaussian Mixture Model (GMM) for pseudo mask creation, as suggested by [42], its performance is still surpassed by the 85% percentile approach. This indicates that the original attention map might contain noise, and a higher threshold is beneficial for generating more effective pseudo masks.

Furthermore, Tab 5d highlights the importance of aggregating multi-head attention maps for mask construction. Given the absence of explicit semantic supervision in the PS pretext task, not aggregating these maps leads to the creation of multiple pseudo masks. This excess of masks introduce ambiguous training objectives for VLP.

**Impact of Mask Refinement.** Refinement of the pseudo masks affects the model's efficacy, as shown in Tab 5f. Performance tends to decrease when either the body mask is omitted or edge smoothing is not applied. However, integrating both these strategies, as we implement in G2D, yields optimal results. This underscores the vital role of pseudo mask refinement in enhancing model performance.

**Ablation on Hyperparameters.** We further ablate the number of attention heads and projector dimensionality. Performance improves with more attention heads, peaking at 3 before slightly declin-

ing at 4 (Tab 5e). Optimal segmentation and classification results are achieved with 128-dimensional projectors. While 256 dimensions provide slight benefits for object detection, they reduce performance in other tasks (Tab 5c). Projectors of 512 dimensions do not yield further gains. Thus, we select 3 attention heads and 128-dimensional projectors for an optimal balance of complexity and effectiveness.

## 5    Conclusion

In this study, we introduce G2D, a novel medical VLP framework for learning global and dense-level representations. Our proposed pixel-level pretext task, pseudo segmentation, leverages a refined attention map to predict a pseudo mask, capturing dense visual features during VLP without requiring additional trainable parameters for its construction. Our model pretrained with this pretext task achieves superior performance across five diverse medical imaging tasks and outperforms methods pretrained with annotated data [5, 21], especially in semantic segmentation. Specifically, on the SIIM [32] dataset, G2D, when fine-tuned with only 1% of the training data, outperforms other medical VLP approaches that utilize the full 100% training set. We anticipate that G2D will inspire further exploration of novel and clinically-guided pretext tasks for medical VLP.

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

# A Appendix / supplemental material

## A.1 Limitations and Future Work

Our work primarily concentrates on learning dense visual representations from pseudo masks, which are generated from attention masks under language supervision. Due to the weak supervision signal, the pseudo masks may not effectively associate each pixel with the corresponding text tokens, potentially capping the performance of our method. Currently, our approach involves learning both global and pixel-level representations through VLP. In future studies, we aim to delve into regional visual representations during VLP to establish more precise correlations between specific chest X-ray (CXR) regions and phrases in radiology reports.

## A.2 Broader Impacts

Our G2D model offers an effective approach for the automatic diagnosis of chest X-ray abnormalities using a small amount of annotated data. This can help decrease the burden on radiologists and enhance healthcare in underprivileged regions. However, medical data, such as chest X-rays and radiology reports, might include sensitive or potentially harmful information. We strongly advise a thorough examination of the data prior to using our model in real-world applications.

## A.3 Pre-training Implementation Details

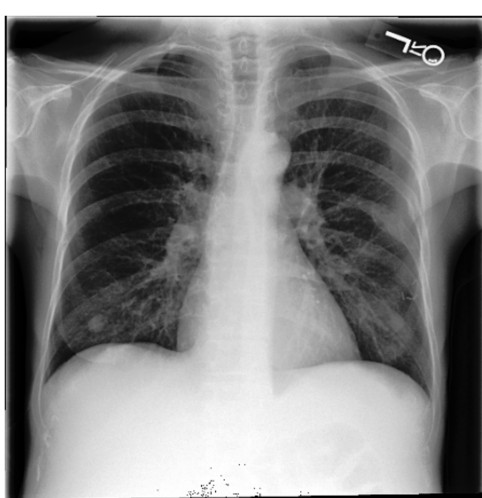

**Report:**
There is no focal consolidation, pleural effusion or pneumothorax. Bilateral nodular opacities that most likely represent nipple shadows. The cardiomediastinal silhouette is normal. The imaged upper abdomen is unremarkable. Chronic deformity of the posterior left sixth and seventh ribs are noted. No acute cardiopulmonary process.

Figure 3: An exemplar pair of X-ray image and associated clinical report from the MIMIC-CXR dataset [24].

The chest X-ray (CXR) images from the MIMIC-CXR dataset [29] are resized to dimensions of $256 \times 256$ and subsequently center-cropped to $224 \times 224$, adhering to the procedure described in [4, 8, 9], with an example shown in Fig 3. The intensity of each image is normalized to a range of $[0, 1]$. During the pre-training stage, we employ data augmentation techniques including random grayscale, random perspective, and auto contrast adjustments, using the PyTorch vision library[1].

## A.4 Downstream Task Implementation Details

The data split information into train/valid/test sets are described in Tab. 6. For all downstream tasks, except from zero-shot image classification and visual grounding, we train with $1\%, 10\%, 100\%$ of the training set. The downstream tasks are deployed on a 40G A100 GPU.

---

[1]https://pytorch.org/vision/stable/transforms.html

Table 6: Details on Data Split: The symbol '/' denotes that training/validation data is not required for the zero-shot tasks.

| Task | Dataset | Split | Train | Valid | Test |
|---|---|---|---|---|---|
| Linear Classification | CheXpert [35] | [35] | 186,027 | 5,000 | 202 |
| | RSNA [31] | [9, 31] | 16,010 | 5,337 | 5,337 |
| | COVIDx [37] | [9, 37] | 23988 | 5998 | 400 |
| Fine-tuned Classification | CXR14 [36] | [21] | 77,872 | 8,652 | 25,596 |
| Image Segmentation | RSNA [31] | [8, 9] | 16,010 | 5,337 | 5,337 |
| | SIIM [32] | [8, 9] | 8,433 | 1,807 | 1,807 |
| Object Detection | RSNA [31] | [8, 9] | 16,010 | 5,337 | 5,337 |
| | Object-CXR [33] | [9] | 6,400 | 1,600 | 1,000 |
| Zero-shot Image Classification | RSNA [31] | [5] | / | / | 5,337 |
| | SIIM [32] | [5] | / | / | 1,807 |
| | CXR14 [36] | [36, 21] | / | / | 25,596 |
| | CheXpert [35] | [43, 21] | / | / | 500 |
| Zero-shot Visual Grounding | RSNA [31] | [5] | / | / | 5,337 |
| | SIIM [32] | [5] | / | / | 1,807 |

### A.4.1 Visual Localization

**Medical Image Segmentation.** For the segmentation tasks on the RSNA [31] and SIIM [32] datasets, we initially employ the vision encoder from the pre-trained model. Additionally, we transfer both the vision encoder and decoder from the pre-trained model, and proceed to train the segmentation network. We implement early stopping during the training process, limiting it to 50 epochs. A learning rate of 2e-4 and a weight decay of 0.05 are adopted. AdamW is utilized as the optimizer, with $\beta_1$ and $\beta_2$ values set at 0.9 and 0.999, respectively. For the SIIM [32] dataset, the default batch size is set at 8, while for the RSNA [31] dataset, it is set at 16. All configurations strictly adhere to the protocol provided in [9].

**Medical Image Object Detection.** The pneumonia detection task on RSNA [31] and foreign objects detection task on Object-CXR [33] datasets are executed on a single A100 GPU. For both datasets, early stopping is implemented during the training process, limited to 50 epochs. AdamW is employed as the optimizer across both datasets. For the RSNA [31] dataset, a batch size of 8 is set for 1% data fine-tuning with a learning rate of 2e-4, a weight decay of 1e-6, and $\beta_1$, $\beta_2$ values of 0.9 and 0.999, respectively. For 10% and 100% data fine-tuning, the batch size is adjusted to 16, with a learning rate of 5e-4, a weight decay of 1e-6, and the same $\beta_1$, $\beta_2$ values. Similarly, for the Object-CXR [33] dataset, a batch size of 8 is set for 1% data fine-tuning, with the identical learning rate, weight decay, and $\beta$ values as the RSNA dataset. For 10% and 100% data fine-tuning, the batch size is adjusted to 16, again with a learning rate of 5e-4, a weight decay of 1e-6, and $\beta_1$, $\beta_2$ values of 0.9 and 0.999. The IOU and NMS thresholds are set at [0.4, 0.45, 0.5, 0.55, 0.6, 0.65, 0.7, 0.75] and 0.5, respectively. All configurations are in strict compliance with the protocol delineated in [9].

### A.4.2 Vision-Language Understanding

**Zero-shot Image Classification.** The original CXR image goes through a two-step preprocessing routine. Initially, it is resized to the dimension of $256 \times 256$, and then center cropped to $224 \times 224$. Following the methodologies outlined in [8, 9], all pixel values are normalized to the range $[0, 1]$. The resulting resized image is then fed through a visual encoder, followed by a visual projector to generate the image embedding $\hat{\mathbf{v}}_i$. Simultaneously, the prompts are fed into a text encoder to obtain text embeddings $\hat{\mathbf{l}}_i$. The classification evaluation hinges on measuring the cosine similarity between the image and text embeddings for each prompt associated with a specific class. The classification outcome is determined by comparing the cosine similarities. Specifically, if the cosine similarity between the image embedding and the positive prompt (e.g., _disease_) surpasses that between the image embedding and the corresponding negative prompt (e.g., _No disease_), the outcome is deemed positive. Conversely, if the reverse holds true, the outcome is negative. The prompt is designed following [7].

**Zero-shot Visual Grounding.** To execute this task, we adhere to the BioViL pipeline as described in [40]. The visual grounding task can be regarded as a pixel-level classification task, driven by the

text prompt and the dense visual embedding. The image is fed into the visual encoder to acquire the dense feature map $\mathbf{V}_i$ from the final convolutional layer of the image encoder, yielding a shape of $C \times H \times W$. At the same time, the prompt is processed through the text encoder and projected into the cross-modal space, resulting in $\hat{\mathbf{l}}_i$. The cosine similarity between $\hat{\mathbf{l}}_i$ and all elements of $\mathbf{V}_i$ at the channel level generates a similarity map. This map is then resized to match the original image size and utilized as the segmentation results to evaluate the zero-shot grounding performance.

### A.4.3 Visual Recognition

We conduct evaluations on the CheXpert [35], RSNA [31], COVIDx [37], and CXR14 datasets [36]. In alignment with previous studies [8, 4, 9, 5, 21], linear classification is implemented on CheXpert [35], RSNA [31], and COVIDx [37]. Here, we update a randomly initialized linear layer while keeping the visual encoder frozen. We adhere to the official test set partition from [5, 21, 36] for a fair comparison. During our linear classification task, training is performed over 50 epochs with a learning rate of 5e-4, a batch size of 8, employing the AdamW optimizer with parameters: $\beta_1 = 0.9$ and $\beta_2 = 0.999$. For the CXR14 dataset [36], we follow the experimental setup from [21], employing fine-tuned classification while updating all parameters from the visual encoder and linear layer. Images are resized to $256 \times 256$ and data augmentation is carried out as recommended in [21]. The AdamW optimizer is utilized with a learning rate of $1 \times 10^{-4}$ and a batch size of 64 for 50 epochs. The linear classification tasks are executed on a single A100 GPU with 40GB memory, using the vision encoder from our pre-trained model as the visual backbone. Fine-tuning is carried out on the randomly initialized linear layer for 50 epochs with early stopping, maintaining a learning rate of 5e-4 and a default batch size of 8. We set AdamW as our optimizer, with $\beta_1$ of 0.9, $\beta_2$ of 0.999, and a weight decay rate of 1e-6.

### A.5 Comparison under MedKLIP Configuration

Table 7: Performance of CXR14 Classification Fine-Tuning and Segmentation Results on SIIM and RSNA using the MedKLIP Setting [5].

|  | CXR14 (AUC) | | | RSNA (Dice) | | | SIIM (Dice) | | |
|---|---|---|---|---|---|---|---|---|---|
|  | 1% | 10% | 100% | 1% | 10% | 100% | 1% | 10% | 100% |
| MedKLIP* | 77.2 | 78.9 | 83.2 | 70.6 | 71.6 | 75.8 | 66.6 | 72.1 | 79.4 |
| **G2D(Ours)** | **80.4** | **83.8** | **86.1** | **73.8** | **76.1** | **76.5** | **70.6** | **74.5** | **82.3** |

To strictly compare our work with MedKLIP [5], we reimplement G2D for fine-tuning on CXR14 classification, as well as SIIM and RSNA segmentation tasks, adhering strictly to the MedKLIP configuration. This approach is necessary because the settings of MedKLIP differ significantly from the other methods that we compare to, such as [4, 8, 9, 6, 21]. Specifically, MedKLIP updates both the encoder and decoder during segmentation tasks, whereas the other methods only update the decoder and keep the encoder frozen. Moreover, MedKLIP employs its own customized data split for CXR14 classification, contrasting with KAD [21], which uses the official CXR14 dataset split.

Given these differences, comparing other methods directly under the MedKLIP setting could be seen as unfair. Therefore, we conducted a separate comparison between G2D and MedKLIP using the MedKLIP setting. The results, presented in Tab 7, demonstrate that G2D outperforms MedKLIP across all tasks and data ratios, even within the MedKLIP setting.

### A.6 Verifying Pseudo Segmentation with Semantic Meaning

To investigate whether the improvements in G2D come from learning dense visual features through pseudo segmentation (PS) or from treating PS as a regularization term during pre-training, we perturbed the semantic integrity of pseudo masks by randomly shuffling them on a sample-wise basis (*i.e.,* making images and pseudo masks unpaired). This operation detaches pseudo masks' semantic connection to the original images, ensuring that the PS task does not learn correct semantic information, but still provide regularisation to the segmentation as the pseudo mask is relatively smooth. The results are presented in Table 8. G2D with uncorrupted pseudo masks in PS (ours) significantly outperforms the results from the shuffled alternative (unpaired images and pseudo masks), not only in

| Mask Construction | SIIM Dice | RSNA mAP | CXR14 AUC |
|---|---|---|---|
| Pseudo Mask without Semantic Meaning (shuffled) | 50.9±2.4 | 7.6±1.2 | 63.7±2.1 |
| Pseudo Mask with Semantic Meaning (**Ours**) | **65.6±1.7** | **15.9±0.8** | **79.1±1.2** |

Table 8: Perturbation on Pseudo Masks.

visual localisation task but also in visual recognition task. The improved performance demonstrate that the proposed G2D indeed learns transferable visual features thanks to the semantic information provided by the pseudo masks, rather than merely treating PS as a regularization mechanism.

### A.7 Pseudo Mask Visualization

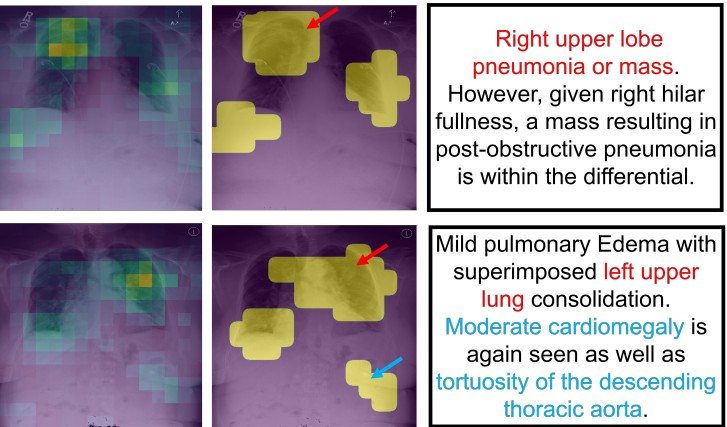

Figure 4: Pseudo Mask Visualization. **Left:** Aggregated attention map. **Middle:** Constructed pseudo mask for the pseudo segmentation task. Red and blue arrows point to areas related to specific text descriptions. **Right:** Corresponding radiology report. Red and blue text emphasize regions represented in the pseudo mask.

We visualize the aggregated attention map, pseudo mask, and paired medical reports in Fig 4. Intriguingly, without human annotations, both the attention map and pseudo mask successfully capture image regions corresponding to various report words. The pseudo masks manage to capture important parts of the image regions related to the highlighted words in the clinical reports, as indicated by the red and blue arrows in Fig 4. This suggests that the supervision signal for the PS pretext task is enriched by the clinical knowledge and high-level semantics, which explain why the PS pretext task may be better than the pixel reconstruction pretext task.

