# OpenReview forum: "G2D: From Global to Dense Radiography Representation Learning via Vision-Language Pre-training"
_NeurIPS.cc/2024/Conference — NeurIPS 2024 poster_

### Official Review · Reviewer_t3s2 · 2024-07-03

**Soundness:** 3
**Presentation:** 3
**Contribution:** 2
**Rating:** 5
**Confidence:** 5

**Summary:**

This paper proposes G2D, a novel vision-language pre-training (VLP) framework for medical imaging that aims to learn both global and dense visual representations from radiography images and their associated radiology reports. The key innovation is a pretext task called Pseudo Segmentation (PS), which uses a pseudo mask derived from attention maps to guide the learning of dense visual features during pre-training. The authors demonstrate that G2D outperforms existing medical VLP approaches on various downstream tasks including classification, segmentation, object detection, and zero-shot visual grounding across multiple medical imaging datasets. Notably, G2D shows strong performance on segmentation tasks even when fine-tuned on very limited data.

**Strengths:**

Novel approach: The paper introduces an innovative method for learning dense visual representations in medical VLP without requiring pixel-level annotations, addressing a key limitation of existing approaches.

Well-motivated: The authors provide a clear rationale for why learning dense representations is important for medical imaging tasks and why existing VLP methods struggle with this.

Comprehensive evaluation: The method is evaluated on a wide range of downstream tasks and datasets, demonstrating its versatility and effectiveness across different medical imaging applications.

Strong results: G2D consistently outperforms existing methods, especially on segmentation tasks where it achieves impressive results with very limited fine-tuning data.

Ablation studies: The paper includes thorough ablation experiments to validate key design choices and components of the method.

Potential impact: The proposed approach could significantly reduce the need for large annotated datasets in medical imaging, which is a major bottleneck in the field.

**Weaknesses:**

Limited theoretical analysis: While the method is empirically strong, there is little theoretical justification for why the pseudo segmentation task leads to improved dense representations.

Complexity of the approach: The method involves several components and processing steps, which may make it challenging to implement and potentially limit its adoption.

Computational resources: The pre-training process appears to be computationally intensive (16 A100 GPUs), which could be a barrier for researchers with limited resources.

Generalization to other domains: While the focus on medical imaging is valuable, it's unclear how well this approach would generalize to other vision-language domains.

Comparison to more recent baselines: Some of the baselines used for comparison (e.g., ConVIRT, GLoRIA) are somewhat older.

Comparison to more recent medical VLP methods would strengthen the evaluation.

**Questions:**

Major concerns:
My primary concern revolves around the authors' claim that current medical VLP methods primarily align images with entire text reports. This assertion appears to be inconsistent with the facts, as evidenced by several papers that have employed local alignment between image regions and text. This factual contradiction significantly undermines the novelty of the present work. For instance:

GLoRIA (Huang et al., ICCV 2021): "Global-Local Representation Alignment for Improved Visual Recognition in Medical Imaging"
This paper introduced a global-local alignment approach, learning finer-grained representations by aligning image patches with text tokens.
MGCA (Wang et al., arXiv 2022): "Multi-Granularity Cross-Modal Alignment for Generalized Medical Visual Representation Learning"
This method employed a multi-granularity alignment strategy, including global, local, and fine-grained levels of alignment.
BioViL (Boecking et al., ECCV 2022): "Making the Most of Text Semantics to Improve Biomedical Vision–Language Processing"
This work proposed a method to improve biomedical vision-language processing by leveraging text semantics, which includes local alignment strategies.
MedKLIP (Wu et al., medRxiv 2023): "Medical Knowledge Enhanced Language-Image Pre-training"
This approach utilized external knowledge bases to enhance local alignment, achieving more fine-grained image-text matching.
Given these existing works, the authors' characterization of the current state of medical VLP appears inaccurate. This misrepresentation significantly weakens the claimed novelty of their approach. The authors should provide a more accurate description of existing methods and clearly articulate how their approach differs from or improves upon these established local alignment strategies.

Other minor concerns:
Have you explored the quality of the learned representations at different levels of the network? Are there significant differences in the quality of features at different scales?
How sensitive is the method to the choice of threshold used in pseudo mask construction? The ablation shows results for a few values, but is there a principled way to choose this threshold?
Have you investigated the potential of using the pseudo masks generated during pre-training for weakly supervised segmentation tasks?
How does the performance of G2D change as the amount of pre-training data is varied? Is there a clear relationship between pre-training data volume and downstream task performance?
Given the computational requirements for pre-training, have you explored any techniques for making the approach more efficient, such as progressive training or curriculum learning?

**Limitations:**

The authors provide a brief discussion of limitations in the appendix, acknowledging potential issues with the weak supervision signal from pseudo masks and the need for further research on regional visual representations. They also touch on broader impacts, mentioning both potential benefits for healthcare and risks associated with sensitive medical data. While these discussions are valuable, they could be expanded to provide more specific insights into the limitations of the current approach and potential mitigation strategies for the identified risks.

---

> ### Author Rebuttal · Authors · 2024-08-05
>
> We thank the reviewer for the questions!
> >Theoretical analysis for why pseudo segmentation task leads to improve dense representations
> - Methods like ConVIRT, GLoRIA, BioViL, MedKLIP, and KAD primarily use an image encoder to extract visual features, aligning them with text embeddings through contrastive learning. MedKLIP and KAD further leverage external tools to extract entities from medical reports for entity classification during VLP However, these methods are limited in their ability to learn dense visual features due to the lack of pixel-level supervision, relying instead on reports or entities.
> - Methods like MRM employ pixel-level pretext tasks, such as masked image reconstruction, for the encoder-decoder vision model. However, this task lacks high-level semantics, as it merely aims to reconstruct the original pixel values [1,2].
> - In contrast, G2D employs pseudo segmentation with pixel-level targets, known as pseudo masks, enabling the vision model to learn dense visual features. G2D's encoder-decoder architecture allows the decoder to become familiar with these dense features, ensuring effective feature learning and continuity when applied to downstream segmentation tasks, unlike methods requiring a randomly initialized decoder. By using pseudo masks derived from attention maps, which align visual features with the medical report, G2D ensures that the pseudo mask reflects the report's semantics. This approach enables the G2D encoder-decoder vision model to learn dense visual features with high-level semantics.
>
> > Complexity of the approach: The method involves several components and processing steps, which may make it challenging to implement and potentially limit its adoption.
> - We kindly disagree. Our method is simple yet effective because **(1)** G2D does not require annotations and only needs image-text pairs; **(2)** Our pretraining is a one-stage process, similar to MGCA, and more efficient than MedKLIP and KAD, which require specific entity extraction from medical reports; **(3)** We directly construct the pseudo mask from the attention map, without relying on an external network, unlike PyramidCLIP[3] and GLIPv2[4].
>
> >Computational resources
> - We use 16 A100 GPUs solely to accelerate the pre-training stage. To ensure a fair comparison with existing methods, we calculate the contrastive loss on each device independently, rather than aggregating batches across devices. Consequently, adding more devices does not increase the batch size or substantially affect the quality of pre-training.
> - We have reimplemented the VLP stage on 2 RTX3090 GPUs, using the same computational resources as MGCA. Due to the smaller RAM of the RTX3090 compared to the A100, we set the batch size to 64 on each GPU. The results of this implementation are shown below. As the table indicates, the downstream performance changes only marginally when pretraining on 2 RTX3090 GPUs, demonstrating that G2D is effective even with less computational resources.
>
> | GPUs | Classification(AUC) | Segmentation(Dice) | Detection(mAP) |
> |-------|----------|----------|----------|
> | | CXR14(1%) | SIIM(1%) | ObjectCXR(1%) |
> | 2 RTX3090   |  78.7 | 65.3  |   3.7 |
> | 16 A100s (default)   |  79.1 | 65.6  |   3.8 |
>
> >Generalization to other domains
> - We implemented pretraining on the MIMIC-CXR dataset, following the protocols established by MGCA, MedKLIP, KAD, and others.
> - Our assumption on image-text region alignment is common and straightforward. Therefore, we believe our method holds value for other vision-language domains, such as natural and remote sensing imagery, due to the importance of dense visual representations across these areas. However, this work is primarily focused on the medical domain. We plan to further explore the potential of our method across various domains in future studies.
>
> >Comparison to more recent baselines
> - We compare our method to M-FLAG (MICCAI 2023), MedKLIP (ICCV 2023), and KAD (Nature Communications 2023), aligning with the comparison methods used in KAD's original work to ensure a fair comparison.
>
> >Pseudo masks for weakly-supervised segmentation tasks
> - This is an interesting direction; we thank the reviewer for the insight and will investigate it in the future. We postulate that our method can serve as an effective pre-training technique for weakly supervised segmentation because it leverages pseudo masks to learn dense visual features without needing precise annotations.
>
> >Effect of the amount of pretraining data
> - For a fair comparison, we implemented the VLP on the full MIMIC-CXR dataset, following the protocols of MGCA, MedKLIP, MRM, KAD, and others. We acknowledge that the amount of pretraining data is a critical and interesting research question for medical VLP, and we plan to investigate this in our future work.
>
> >Progressive training or curriculum learning
> - Thank you for the advice. We have conducted an ablation study using less computational resources—2 RTX 3090s. As shown in the table above, the results demonstrate that G2D also performs well with fewer computational resources.
>
>
> >Clarification on other medical VLP methods’ limitations in alignment\
> >Quality of different level visual features\
> >Sensitivity analysis on threshold used in pseudo maks construction
> - We have added a detailed explanation in the official comment below. Please refer to it for more information.
>
> [1] Liu Y, et al. Improving pixel-based mim by reducing wasted modeling capability, ICCV, 2023.\
> [2] Liu Y, et al. PixMIM, TMLR, 2023\
> [3] Gao, Yuting, et al. PyramidCLIP, NeurIPS 2022\
> [4] Zhang, Haotian, et al. GLIPv2, NeurIPS 2022

---

> > ### Comment · Reviewer_t3s2 · 2024-08-08
> > **The rebuttal addresses my concerns**
> >
> > I thank authors for their efforts in addressing my concerns. After reading the rebuttal, my concern has been addressed. And I will update the final rating.

---

> > > ### Author Response · Authors · 2024-08-09
> > >
> > > Thank you for your feedback and for considering our response. We appreciate the opportunity to clarify our work and are grateful for your thoughtful review.

---

> ### Author Response · Authors · 2024-08-05
>
> Continue with the rebuttal
>
> >Clarification on other medical VLP methods’ limitations in alignment
> - We have described GLoRIA, MGCA, MedKLIP, and KAD in Section 2 of the main article and will provide a detailed explanation and comparison with G2D in the following sections.
> - **Ambiguous Token-level Alignment in GLoRIA and MGCA:** Both GLoRIA and MGCA employ a brute-force approach to align image and text tokens. This token-level alignment might compromise the medical context and lead to misalignments. For example, medical terms such as 'compatible' or 'acute' lack direct visual correlates, making local alignment ambiguous.
> - **Global Alignment only in BioViL**: BioViL implements only global alignment, as detailed in their original work (equation 2 and section 2.2). During pre-training, their loss functions include global image-text alignment and masked language modeling on the text side, but they do not incorporate a loss for dense visual representation learning.
> - **Loss Functions in MedKLIP**: During pre-training, MedKLIP utilizes an entity classification loss, applying it to all image features for classifying entities, and a contrastive loss where different positional names are treated as negative samples using their name embeddings for contrastive learning. However, these loss functions are not explicitly designed for fine-grained image-text matching.
> - **Unique Approach of G2D**: Unlike GloRIA, MGCA, BioViL, MedKLIP, and KAD, which only use an image encoder during pretraining, G2D employs an encoder-decoder architecture to enhance visual representation learning. During VLP, the encoder extracts global visual features aligned with text to learn global visual representations. Additionally, G2D incorporates a decoder that performs pseudo-segmentation tasks using pseudo masks generated by the G2D encoder, independent of external annotations. This decoder leverages features from the image encoder for pseudo-segmentation, enabling both the encoder and decoder to jointly learn dense visual representations.
>
> >Quality of different level visual features
> - Most network structures for dense prediction tasks use an encoder-decoder architecture, where the features at the **penultimate layer of the decoder** are used for final pixel-wise prediction. For image classification, the structure usually involves an encoder with a linear classifier, where the features of the **penultimate layer of the encoder** are used for prediction. For both **representative cases**, the effectiveness of the penultimate layer of the encoder or decoder is demonstrated in Section 4. These experiments have shown the effectiveness of our approach in the most general settings (i.e., both classification and dense prediction).
>
> >Sensitivity analysis on threshold used in pseudo maks construction
> - We conducted a detailed ablation study on various threshold values when building the pseudo masks. The results are shown below. As the table indicates, the best downstream performance is achieved with an 85th percentile threshold. Increasing the threshold to the 95th percentile does not improve performance, suggesting that an extremely high threshold may lead to over-filtering. Conversely, decreasing the threshold from the 85th to the 25th percentile consistently degrades performance, as a lower threshold causes the pseudo mask to cover most of the image, introducing noise during VLP. Based on these experimental results, we empirically set our threshold to the 85th percentile.
> - In the future, we will investigate using an adaptive threshold to filter the attention map.
>
> | Threshold | Classification(AUC) | Segmentation(Dice) | Detection(mAP) |
> |-------|----------|----------|----------|
> | | CXR14(1%) | SIIM(1%) | ObjectCXR(1%) |
> | 95% percentile   |  78.5 | 64.8  |   3.7 |
> | 85% percentile (default) |  79.1 | 65.6  |   3.8 |
> | 75% percentile |   78.3 | 63.0  |   3.4 |
> | 50% percentile (median) |   75.6 | 58.8  |   2.3 |
> | 25% percentile |  75.2 | 65.6  |   2.1 |

---

### Official Review · Reviewer_SZ6L · 2024-07-08

**Soundness:** 3
**Presentation:** 3
**Contribution:** 3
**Rating:** 5
**Confidence:** 4

**Summary:**

This manuscript describes a medical vision-language pre-training framework called Global to Dense level representation learning (G2D), that learns global and dense visual features simultaneously with only image-text pairs, by exploiting the aggregated attention map from the vision encoder for a pseudo segmentation pretext task. The improved (frozen) vision encoder is then utilized as part of the model pipeline for a number of downstream tasks (e.g. segmentation, classification)

**Strengths:**

- Pseudo segmentation pretext task enables dense segmentation during pre-training, and avoids external resources as for alignment-based methods, and limitations on high-level semantic representations in reconstruction-based methods
 - Importance of associating semantic meaning verified via experiment

**Weaknesses:**

- Unclear if specific sentence/phrase to individual image region alignment is achieved, for dense learning
 - Lack of fine-grained pixel-level evaluation of masks

**Questions:**

1. The accuracy of the initial aggregated attention map appears possibly non-optimal, given that additional thresholding by body mask is required. As such, it might be considered to quantify the accuracy of these maps, possibly against segmentation ground truth.

2. In Section 3.2, it is stated that a threshold is applied (at 85%) to transform the aggregated attention map into a binary mask, before smoothing. It might be clarified if the need for smoothing (and related smoothing parameters) was empirically determined.

3. In Section 3.3, it is stated that "This decoder takes visual feature V_i as input and utilises the pseudo mask ˜M_i as the supervisory signal for the pretext task". It might be clarified as to whether and how specific text can be matched to specific (separate) image regions, as in Figure 4 of Section A.7. In other words, while Figure 4 shows specific text descriptions corresponding to specific image regions, were these correspondences/alignments indicated by the proposed G2D model, or are they external manual observations? A.1 suggests no, but this might be explicitly stated.

4. In Section 4, the choice of ResNet-50 as the encoder over other plausible choices (e.g. U-Net encoder) might be briefly explained.

5. For Table 1, it might be clarified as to what "encoder-decoder" refers to - the updating of both encoder and decoder?

---

> ### Author Rebuttal · Authors · 2024-08-05
>
> We thank the reviewer for the positive feedbacks!
> > Unclear if specific sentence/phrase to individual image region alignment is achieved, for dense learning (W1)
> - Since the MIMIC-CXR pretraining dataset does not establish a direct relationship between specific sentences or phrases and image regions, direct alignment between them is infeasible. To facilitate the learning of dense visual features, we construct pseudo masks for a pseudo segmentation pretext task, as detailed in Section 3.2.
> - Notably, our method has a significant advantage during medical VLP as it learns dense visual features using only image-text pairs. It does not rely on specific sentence or phrase annotations paired with individual image regions. This means our approach is more generalizable and does not require region-level annotations.
>
> > Evaluation of pseudo masks. (W2, Q1)
> - In the G2D approach, our objective is to enhance medical VLP by designing a pseudo mask for learning dense visual features through a pseudo segmentation task. This method does not rely on guessing the semantic mask as in traditional supervised learning. Since the MIMIC-CXR dataset used for pretraining **lacks pixel-level annotations**, it is impossible to directly assess the accuracy of the pseudo masks created through G2D. However, preliminary quality checks, as detailed in Appendix A.7 and illustrated with two examples in Figure 4, show that G2D successfully identifies image regions that align with the content of the entire report using just language cues and pseudo mask supervision.
> - We acknowledge that the pseudo masks are not perfect. However, adding body masks is one of the simplest and most commonly used operations in medical image processing[1].
>
> > Ablation on smoothing pseudo masks (Q2)
> - We assess the impact of smoothing on the G2D model's performance in Table 5-f by comparing versions with and without smoothing. The results show that the model variant with smoothing outperforms the one without, especially in the SIIM segmentation task. This indicates that smoothing the pseudo mask enhances the learning of representative visual features during VLP particularly for dense visual features.
> - To further investigate the effect of smoothing, we performed an ablation study on the 'window_size' parameter when implementing a bilateral filter. The results are shown below. As the findings indicate, variations in 'window_size' do not substantially affect the quality of VLP, demonstrating that the G2D method is robust across different parameter settings for smoothing operations.
>
> | Window Size | Classification(AUC) | Segmentation(Dice) | Detection(mAP) |
> |-------|----------|----------|----------|
> | | CXR14(1%) | SIIM(1%) | ObjectCXR(1%) |
> | 5$\times$5   |  79.2 | 65.5  |   3.8 |
> | 7$\times$7 (ours)   |  79.1 | 65.6  |   3.8 |
> | 10$\times$10   |  79.1 | 65.6  |   3.7 |
>
> > Clarifying Text-to-Image Region Correspondences in G2D Model (Q3)
> - During pre-training, the pseudo mask for each sample is generated by the G2D model using the attention map, without the need for external manual annotations.
> - Our approach does not attempt to align specific text with specific image regions. Instead, our primary objective is to establish a semantically meaningful target for dense representation learning during pre-training. We utilize the semantic information encoded in the image-text attentions, rather than focusing on precisely predicting downstream segmentation tasks, which is theoretically unrealistic.
> - In Appendix A.7, the image region depicted is the pseudo mask derived from the G2D model, not one annotated by humans.
> - We will clarify this part in the camera-ready version according to your suggestion.
>
> > Choice of U-Net encoder (Q4)
> - ResNet-50 is the most commonly used vision encoder for vision-language pre-training (VLP) methods, such as CLIP.
> - For a fair comparison, we strictly adhere to the protocols established by MGCA, MedKIP, KAD, and others, using ResNet-50 as the image encoder.
> - The latest version of nnU-Net [4] utilizes a ResNet backbone and demonstrates improved performance compared to a vanilla, non-ResNet backbone.
>
> > Clarification for Tab 1 'encoder-decoder' (Q5)
> - We adhere to the protocols established by GloRIA[2] and MGCA[3] by only updating the decoder while keeping the encoder frozen during training. We will update and clarify this part in the camera-ready version of our document.
>
> [1] Imura, Masataka, et al. "Automatic cropping method of chest radiographs based on adaptive binarization." EMBC, 2013.\
> [2] Huang, Shih-Cheng, et al. "Gloria: A multimodal global-local representation learning framework for label-efficient medical image recognition." ICCV. 2021\
> [3] Wang, Fuying, et al. "Multi-granularity cross-modal alignment for generalized medical visual representation learning." NeurIPS 2022\
> [4] Isensee, Fabian, et al. "nnU-Net Revisited: A Call for Rigorous Validation in 3D Medical Image Segmentation." CoRR 2024.

---

> > ### Comment · Reviewer_SZ6L · 2024-08-12
> >
> > We thank the authors for their clarifications.

---

### Official Review · Reviewer_bXf3 · 2024-07-09

**Soundness:** 4
**Presentation:** 4
**Contribution:** 3
**Rating:** 6
**Confidence:** 4

**Summary:**

The paper proposes an encoder-decoder medical VLP approach for global-to-dense visual representation learning. Pseudo segmentation is adopted for dense level learning. Rich experiments validate the effectiveness of the proposed method.

**Strengths:**

1. The motivation behind the work is clear. Pseudo-segmentation supervision is effective, which is validated by experiments.
2. The experiments are rich and ablation analysis shows the contributions of each component and design.
3. The illustrations are clear and easy to understand.
4. The improvements are consistent and sometimes substantial.

**Weaknesses:**

1. The comparisons with MGCA and MRM in the CXR14 dataset are not included in Table 3, but Table 4 includes the comparisons with MGCA and MRM. What are the reasons behind this?
2. Transformer-based vision encoder is not analyzed.
3. The balance between VLA and PA losses is not analyzed.

**Questions:**

Is it not applicable to compare with MGCA and MRM in the CXR14 dataset?

---

> ### Author Rebuttal · Authors · 2024-08-05
>
> We thank the reviewer for the positive feedbacks!
> >Comparing with MGCA and MRM on CXR14 datasets (W1,Q1)
> - In Table 3, we directly reference the results from the KAD study to ensure a fair comparison, as KAD[1] uses the official data split for CXR14. It's important to note that the KAD[1] study does not include results for MGCA[2] and MRM[3].
> - In the MGCA[2] study, results on CXR14 are not reported, which makes direct comparison challenging. Meanwhile, the MRM study, although it includes experiments on CXR14, uses its own data split rather than the official split provided by CXR14. This leads to potential biases when comparing it with existing methods with official split， such as KAD. Therefore, in Table 3, we only compare our results with those methods that have reported outcomes using the official CXR14 split, as documented by the KAD study.
> - To comprehensively compare our G2D method with MGCA and MRM, we re-implemented both on CXR14 using the official split as employed by KAD. For finetuning, we utilized their officially provided pretrained weights. Additionally, we used the official finetuning code from the MRM GitHub repository, making a single modification: we replaced their own data split with the official split used by KAD. The results are shown in the table below. As the table indicates, G2D substantially outperforms both MGCA and MRM on the CXR14 dataset using the official data split:
> | Methods | CXR14(1%) | CXR14(10%) | CXR14(100%) |
> |-------|----------|----------|----------|
> | MGCA[2]   | 62.4  |  73.9 |  81.2 |
> | MRM[3]     | 64.2  |  74.3 |  81.0 |
> | **G2D(ours)**   | **79.1**  | **81.1**  | **83.1**  |
>
> >Transformer-based vision encoder is not analyzed (W2)
> - We conducted the experiments using a transformer-based vision encoder, specifically the ViT, configured identically to that used in the MRM study [3]. For the vision decoder, we also employed a transformer-based architecture, the same as MRM. The results of G2D with various transformer variants are shown in the table below. As indicated by the table, the performance of G2D does not fluctuate significantly with different backbone types, demonstrating that our method is backbone-agnostic.
>
> | Backbone | Classification(AUC) | Segmentation(Dice) | Detection(mAP) |
> |-------|----------|----------|----------|
> | | CXR14(1%) | SIIM(1%) | ObjectCXR(1%) |
> | G2D(CNN)   |  **79.1** | 65.6  |   **3.8** |
> | G2D(Transformer)   | 78.8  | **65.7**  |  3.5 |
>
> >The balance between VLA and PA losses is not analyzed (W3)
> - We selected a coefficient of 1 for both the VLA and PA losses from the initial development of the project, as we believe a robust method should not require specifically tuned coefficients for each loss.
> - Due to time constraints during rebuttal, we plan to ablate the coefficients of these two losses in future work to comprehensively investigate their contributions to the G2D method.
>
> [1] Zhang, Xiaoman, et al. "Knowledge-enhanced visual-language pre-training on chest radiology images." Nature Communications 2023\
> [2] Wang, Fuying, et al. "Multi-granularity cross-modal alignment for generalized medical visual representation learning." NeurIPS 2022\
> [3] Zhou, Hong-Yu, et al. "Advancing Radiograph Representation Learning with Masked Record Modeling." ICLR 2023.

---

> > ### Comment · Reviewer_bXf3 · 2024-08-09
> > **Thank you**
> >
> > Thank you for addressing my concerns! I have no further questions and maintain my original rating.

---

> > > ### Author Response · Authors · 2024-08-09
> > >
> > > Thank you for taking the time to reassess your concerns and for maintaining your positive feedback. We truly appreciate your thoughtful insights and feedback.

---

### Official Review · Reviewer_iPQX · 2024-07-13

**Soundness:** 3
**Presentation:** 4
**Contribution:** 3
**Rating:** 6
**Confidence:** 4

**Summary:**

The paper proposes a new medical vision-language model, G2D, which employs vision-language alignment (VLA) and pixel alignment (PA) strategies, combined with a pseudo segmentation (PS) pre-training task, to learn global and dense visual representations from medical images. The VLA strategy is used to learn global representations of images and texts, while the PS task constructs pseudo masks through a parameter-free mechanism to facilitate the learning of dense representations. The method is comprehensively validated across five downstream tasks (image segmentation, object detection, zero-shot image visual grounding, zero-shot image classification, and fine-tuned image classification), demonstrating its effectiveness in handling both unimodal and cross-modal tasks.

**Strengths:**

+ The paper is well-written, with the motivation, method, and results clearly presented. A minor concern is the reference format; it should be [1] instead of (1) according to the NeurIPS template.

+ A significant concern with most existing works is that they operate primarily at the Image-Text Retrieval level, similar to the perceptual level of CLIP, and do not effectively capture dense features between modalities. The G2D model addresses this issue by integrating Vision-Language Alignment (VLA) and Pseudo Segmentation (PS) tasks to facilitate simultaneous learning of global and dense visual features. This multi-level feature learning significantly enhances the model's performance in tasks requiring dense feature perception, such as segmentation.

+ During pre-training, the G2D method utilizes only image-text pairs without the need for additional annotated data. By generating pseudo masks on the fly through the PS task, it reduces the cost and complexity associated with data annotation.

+ The G2D method is novel, and the experiments are robust. Experimental results on five medical imaging tasks involving 25 diseases demonstrate that the G2D model outperforms existing models, even with minimal fine-tuning data. Notably, in segmentation tasks requiring dense visual features, G2D achieves excellent results with just 1% of the training data for fine-tuning.

**Weaknesses:**

Major concerns:

- The attention maps could introduce errors in pseudo mask, and these errors may propagate throughout the training process. To address this, a clear validation strategy needs to be outlined. For instance, in Figure 2, aggregated attention map might incorrectly highlight irrelevant regions. It is essential to establish methods for **detecting** and **measuring** these errors to ensure the reliability of the model. I hope the authors could quantify the errors in aggregated attention map and pseudo mask during the rebuttal period.

Minor concerns:

- The training and validation of the model rely on specific datasets, which may introduce biases and potentially affect the model's generalizability to different datasets.

- It is uncertain whether the method can be effectively extended to vision-language tasks involving 3D imaging (e.g., CT and MRI), presenting a limitation in its current scope of application.

**Questions:**

- How do you detect and correct the errors made by aggregated attention map?

**Limitations:**

Limitations were discussed in Section A.1

---

> ### Author Rebuttal · Authors · 2024-08-05
>
> We thank the reviewer for the positive feedbacks!
> >Detecting and measuring the error of pseudo mask (W1, Q1)
> - In G2D, we aim to design pseudo mask for learning dense visual feature from pseudo segmentation task during medical vision-langauge pre-training (VLP), rather than directly guess the semantic mask for supervised learning.
> - Since the MIMIC-CXR dataset, which is used for pretraining, does not have segmentation mask annotations, it is infeasible to directly evaluate the accuracy of the pseudo mask derived from the G2D method.
> - In Appendix A.7, where we conducted a quality check on several samples and visualized two examples in Figure 4. We observed that G2D is capable of learning the image regions of interest that correspond with the entire report, using only language and pseudo mask supervision. However, a detailed quantitative evaluation would require laborious work by clinicians, and we plan to consider this in future studies.
>
> >The training and validation of the model rely on specific datasets, which may introduce biases and potentially affect the model's generalizability to different datasets.(W2)
> - We compared our approach with well-established works such as GLoRIA (ICCV 2021), MGCA (NeurIPS 2022), MedKIP (ICCV 2023), and KAD (Nature Communications 2023). All these studies utilize the MIMIC-CXR dataset, which is commonly used for 2D medical VLP.
> - To ensure a fair comparison with existing methods, we strictly adhered to their experimental settings [1,2,3,4], using the same datasets for both pretraining and downstream evaluation.
> - Furthermore, due to limitations in publicly accessible datasets, MIMIC-CXR is the only large-scale medical image-text dataset, containing over 200,000 samples, available for implementing medical VLP. We hope the research community will release more publicly available datasets for VLP to reduce bias and enhance model generalizability.
>
> >It is uncertain whether the method can be effectively extended to vision-language tasks involving 3D imaging (e.g., CT and MRI), presenting a limitation in its current scope of application. (W3)
> - Our method can be easily adapted to 3D imaging modalities by replacing the 2D image encoder with a 3D version. However, there is currently no public large-scale 3D image-text dataset comparable to MIMIC-CXR, which has over 200,000 samples, for implementing 3D medical VLP. We note that scaling our proposed framework to native 3D is straightforward because the pseudo masks are derived from attentions (see Section 3.2). We will explore the potential of our work further if such datasets become publicly accessible.
>
> [1] Huang, Shih-Cheng, et al. "Gloria: A multimodal global-local representation learning framework for label-efficient medical image recognition." ICCV. 2021\
> [2] Wang, Fuying, et al. "Multi-granularity cross-modal alignment for generalized medical visual representation learning." NeurIPS 2022\
> [3] Wu, Chaoyi, et al. "Medklip: Medical knowledge enhanced language-image pre-training for x-ray diagnosis." ICCV 2023\
> [4] Zhang, Xiaoman, et al. "Knowledge-enhanced visual-language pre-training on chest radiology images." Nature Communications 2023

---

> > ### Comment · Reviewer_iPQX · 2024-08-09
> >
> > I appreciate the authors' detailed responses, which effectively addressed my previous concerns. As a result, I'd like to raise my rating to Weak Accept.
> >
> > Regarding pseudo label evaluation, per-voxel annotations may not be necessary. Based on the report, if the disease is present in the image and the pseudo labels correctly identify it, this counts as a true positive; otherwise, it's a false negative. Similarly, if the report indicates the image is healthy, the authors could calculate the number of true negatives and false positives for the pseudo labels.
> >
> > This strategy might be able to evaluate the quality of pseudo labels.

---

> > > ### Author Response · Authors · 2024-08-11
> > >
> > > Thank you for your thoughtful review and for considering an upgrade in your rating based on our responses. We are grateful for your suggestion on evaluating pseudo labels without per-voxel annotations and will explore implementing this strategy to further validate our methodology. Your insights are invaluable to enhancing the quality of our work.

---

### Decision · Program_Chairs · 2024-09-25

**Decision:**

Accept (poster)

**Comment:**

The paper addresses an interesting problem of learning global and dense visual features via vision-language alignment (VLA) and pixel alignment (PA) strategies. The proposed method demonstrates its superior performance on various medical tasks, i.e., image segmentation, object detection, visual grounding, and image classification (both zero-shot and fine-tuned). The reviewers all agree that the paper is well-written and the proposed pseudo-segmentation pretext task of exploiting the aggregated attention maps from the vision encoder is novel and valuable.  In the initial review comments, the reviewers raised concerns about the quality of pseudo masks, how they will affect the training, the feasibility of being extended to perform in 3D, and some details of other techniques. The authors provided a detailed response and supportive experimental results, and most of the concerns were addressed, resulting in favorable final ratings from the reviewers. Overall, the paper introduced a novel pre-training scheme, promoting dense semantic information in medical images, and demonstrated the superior performance gain of the proposed method, which AC believes could benefit and inspire the related field and thus worth accepting to NeurIPS. The AC encourages the authors to further address the review comments and strengthen the paper.